# FORGE: Forward-Only Test-Time Adaptation for Integer-Only Vision Models on Microcontrollers

## Abstract

Vision models deployed on microcontrollers (MCUs) are quantized to integer-only arithmetic and run in inference-only runtimes that do not carry the machinery backpropagation needs: the standard tool for adapting a model to the distribution shift (sensor noise, blur, lighting) it meets in the field. Existing forward-only test-time adaptation (TTA) methods either run only on server- or edge-GPU-class models (not true microcontroller integer execution), or require the batch-normalization (BN) layers that integer deployment fuses away. We present a forward-only TTA method that operates on *deployed*, BN-folded, integer-only convolutional networks. The key observation is that fusing BN into the preceding convolution, a mandatory step for integer inference, destroys the statistics that normalization-based adaptation relies on. We restore adaptation by re-normalizing each folded convolution's per-channel output to its clean training statistics, using only forward-pass estimates. The method (i) recovers most of gradient-based TENT's accuracy gain (+20.9 vs. +24.9 points) and matches forward-only BN adaptation, while being the only method that runs on a folded integer-only model; (ii) needs to adapt only 3 of 21 layers (selected without seeing the test corruptions) to recover 93% of the benefit; (iii) survives single-sample streaming with a batch-size-scaled momentum; and (iv) generalizes across three datasets (up to 200 classes) and two architectures. We validate true integer-only execution and deploy on an ESP32-S3, where, measured with a Nordic PPK2 power profiler, adaptation costs only **8.3 mJ (6.8% of inference energy)** and **21.9 ms** on the deployed SIMD-optimized model: forward-only adaptation is cheap on a real microcontroller.

## 1 Introduction

On-device vision on microcontrollers is attractive for privacy, latency, and energy: no data leaves the device and no network is required. To fit kilobytes of SRAM and run on integer-only arithmetic units, models are quantized to int8 and their batch normalization (BN) layers are *folded* into the preceding convolutions. This is mandatory for efficient integer inference (TFLite-Micro, CMSIS-NN, ESP-NN), but it has a consequence that is rarely discussed: the deployed model can no longer be adapted.

Models degrade under distribution shift (noise, blur, weather, compression) that they never saw during training (Hendrycks & Dietterich, 2019). On a server, one simply fine-tunes or runs test-time adaptation (TTA). On the deployed MCU model this is out of reach for two compounding reasons. First, the deployed inference-only runtime does not carry the machinery backpropagation needs (an autograd graph, fp32 master weights, optimizer state), so gradient-based TTA such as TENT (Wang et al., 2021) or CoTTA (Wang et al., 2022) does not run on it as deployed; on-device training frameworks *can* backpropagate through quantized models (Lin et al., 2022; Deutel et al., 2024; Buron et al., 2025), but only by reintroducing that machinery at substantial memory and compute cost (Sec. 2). Second, the forward-only alternative, recalibrating BN statistics (Schneider et al., 2020; Dong et al., 2025), requires BN layers, which folding has removed. The deployed model thus degrades exactly as much as its float counterpart, but *none* of the existing remedies apply to it.

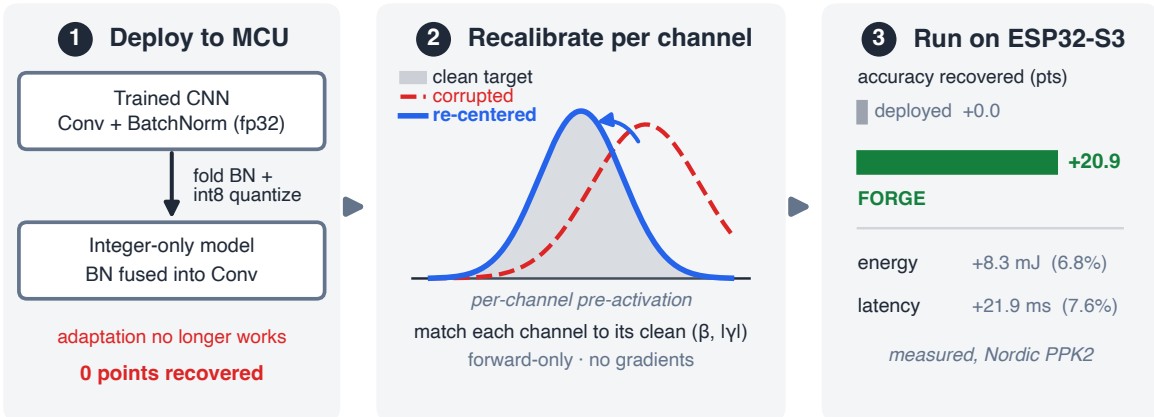

Figure 1: **Forge in one view. (1)** Deploying a CNN to a microcontroller folds batch normalization into the preceding convolutions and quantizes to int8; the normalization statistics that test-time adaptation relies on are baked in, so adaptation out of the box recovers nothing (+0.0). **(2)** FORGE restores adaptation *forward-only*: under corruption each folded channel's output distribution drifts (red), and FORGE re-centers it onto its clean training target $(\beta, |\gamma|)$ (blue) from running per-channel statistics, using no gradients. **(3)** On an ESP32-S3 this recovers +20.9 accuracy points at a measured 6.8% extra energy (8.3 mJ) and 21.9 ms on ESP-NN SIMD kernels, profiled with a Nordic PPK2.

Recent forward-only TTA methods do not close this gap. FOA (Niu et al., 2024), ZOA (Deng et al., 2025), and PACE (Sójka et al., 2026) are gradient-free but are demonstrated on server- and edge-GPU-class hardware, not under true microcontroller integer execution; PEA (Ma et al., 2026) is gradient-free but does not target quantization; and TinyTTA (Jia et al., 2024) runs on an MCU but is gradient-based (it only reduces backprop memory). None reports the one quantity that matters for a battery device: measured on-device energy.

We close this gap. Our contributions are:

- We identify and empirically demonstrate the *adaptation gap*: folding BN for integer deployment removes the statistics that forward-only normalization adaptation needs, eliminating its +20-point recovery (Sec. 3.2).

- We propose FORGE (Forward-Only Recalibration for the edGE), a forward-only per-channel recalibration that restores adaptation on the deployed folded integer-only model, recovering most of gradient-based TENT's gain and matching forward-only BN adaptation, while being uniquely deployable (Sec. 3, Table 2).

- We show adaptation is needed at only a few layers: 3 of 21 layers, selected on held-out corruptions, recover 93% of the benefit, and adapting the right subset *exceeds* adapting all layers (Sec. 3.3).

- We characterize single-sample streaming and show a batch-size-scaled momentum rescues it; and we demonstrate generalization across three datasets (CIFAR-10, CIFAR-100, and Tiny-ImageNet, spanning up to 200 classes) and two architectures.

- We validate true integer-only execution and deploy on an ESP32-S3, reporting the first *measured* energy cost of forward-only TTA on an MCU: 8.3 mJ (6.8% of inference energy), 21.9 ms, on ESP-NN SIMD kernels (Sec. 4.7).

A central message is that adapting a deployed integer-only model does not require a complex algorithm; it requires recognizing what folding removes and restoring it cheaply. The adaptation step itself is intentionally

minimal: no gradients, no learnable parameters, and no extra forward passes. That minimalism is precisely what makes it run, at negligible energy, inside a deployed runtime that does not backpropagate. The novelty is in the problem (the deployment gap), the efficiency (only a few layers, single-sample streaming), and the evidence (true integer-only execution with measured energy), not in the adaptation rule.

## 2 Related Work

**Gradient-based TTA.** A large family of test-time adaptation methods updates the model by backpropagation: TENT (Wang et al., 2021) minimizes prediction entropy over the BN affine parameters; SHOT (Liang et al., 2020) uses information maximization and pseudo-labels; EATA (Niu et al., 2022) and SAR (Niu et al., 2023) improve efficiency and stability under non-i.i.d. streams; MEMO (Zhang et al., 2022) enforces augmentation consistency; and CoTTA (Wang et al., 2022) and NOTE (Gong et al., 2022) target the continual setting. These methods define the accuracy ceiling, but the obstacle to deploying them on an MCU is structural rather than a matter of efficiency: backpropagation needs the stored activation graph and optimizer state of a floating-point model, neither of which an integer-only runtime (TFLite-Micro, ESP-NN) materializes. Even memory-reduced variants still differentiate, so they remain off the integer path. We use TENT and CoTTA as (undeployable) accuracy references.

**Forward-only normalization adaptation.** Adapting normalization statistics forward-only is a longstanding idea: AdaBN (Li et al., 2018) replaces source BN statistics with target-domain ones, and He & Cheng (2018) recalibrate BN statistics from limited unlabeled data to restore accuracy after quantization and pruning. More recently, prediction-time BN adaptation (Schneider et al., 2020; Nado et al., 2020) and LeanTTA (Dong et al., 2025) recalibrate normalization statistics at test time without gradients, and T3A (Iwasawa & Matsuo, 2021) adjusts the classifier with a bank of stored prototypes. They are the closest forward-only baselines, and they share a dependence FORGE removes: *all of them assume a live batch-normalization layer to refresh* (AdaBN, He & Cheng, BN-adapt, LeanTTA) or a cached representation of the source feature space (T3A). This is exactly the regime integer deployment destroys: once BN is folded into the convolution for integer-only inference the statistics are gone, and the feature bank is not shipped, so none of these methods has anything to update. FORGE differs precisely by operating *after* the fold: it carries only two per-channel constants per former-BN site, recorded for free at fold time, and recovers the statistics post-fold, which is what lets it run on the deployed model. The mechanism is deliberately close to this prior line of work; the contribution is making it run where the BN layer no longer exists (Sec. 5).

**Forward-only TTA for quantized models.** FOA (Niu et al., 2024) and PACE (Sójka et al., 2026) adapt quantized ViTs with derivative-free optimization (CMA-ES); ZOA (Deng et al., 2025) uses zeroth-order estimation and covers CNNs. All three run on GPU or edge-GPU hardware (PACE's only on-device number is a Jetson Xavier NX throughput), and none reports MCU energy. PEA (Ma et al., 2026) is forward-only and architecture-agnostic but does not address quantization.

**TTA on microcontrollers.** TinyTTA (Jia et al., 2024) is the closest prior work and the only other TTA method validated on an MCU (STM32H747), but it is gradient-based: a self-ensemble + early-exit strategy that *reduces* backprop memory rather than eliminating gradients, and it adapts the network while its normalization layers are still present. On the deployed *folded* integer-only model it therefore falls in the same bucket as TENT and BN-adapt (Table 1): it needs the gradients and the live normalization layers that folding removes, so it does not run on the deployed model as-is and a like-for-like accuracy row would not be apples-to-apples. FORGE differs on every axis that matters for deployment: forward-only, true integer-only, on a folded model, with measured energy. The comparison below makes this quantitative: the two methods solve the same problem (small-batch TTA on microcontrollers) from opposite ends, TinyTTA keeping gradients (and adding trained early-exit heads) before the fold, FORGE forward-only on the deployed folded int8 model. Entries are from the respective papers (Jia et al., 2024 for TinyTTA, this work for FORGE).

Table 1: What it takes to adapt a deployed integer-only MCU model. "int-only" = true integer execution (not simulated); "energy" = measured on-device energy. Each prior method satisfies only a subset.

| method | fwd-only | no BN | int-only | MCU dep. | meas. energy |
|---|---|---|---|---|---|
| TENT (Wang et al., 2021), CoTTA (Wang et al., 2022) | ✗ | ✗ | ✗ | ✗ | ✗ |
| BN-adapt (Schneider et al., 2020) | ✓ | ✗ | ✗ | ✗ | ✗ |
| LeanTTA (Dong et al., 2025) | ✓ | ✗ | ✗ | ✗ | ✗ |
| FOA/ZOA/PACE (Niu et al., 2024; Deng et al., 2025; Sójka et al., 2026) | ✓ | ✓ | ✗ | ✗ | ✗ |
| PEA (Ma et al., 2026) | ✓ | ✓ | ✗ | ✗ | ✗ |
| TinyTTA (Jia et al., 2024) | ✗ | ✗ | ✗ | ✓ | ✗ |
| **Forge (ours)** | ✓ | ✓ | ✓ | ✓ | ✓ |

**Forge vs. TinyTTA, the closest prior MCU TTA method.**

| property | TinyTTA (Jia et al., 2024) | FORGE (ours) |
|---|---|---|
| adaptation mechanism | gradient-based early-exit ensemble | forward-only recalibration |
| gradients / optimizer state | yes (reduced via early exit) | none |
| normalization layers | trained early-exit heads, pre-fold | post-fold $(\beta, |\gamma|)$ |
| runs on deployed folded int8 model | no | yes |
| batch size 1 | yes | yes (window-matched momentum) |
| adaptation memory | early-exit heads + backprop buffers | 784 fp32 scalars ($\approx 6$ KB) |
| measured on-device energy | not reported | 8.3 mJ (6.8%) |
| hardware validated | STM32H747 (Cortex-M7) | ESP32-S3 |

**On-device training of quantized models.** A separate line of work backpropagates through quantized networks *on the device*: MCUNet's tiny training engine fits training into 256 KB via sparse layer updates and quantization-aware scaling (Lin et al., 2022), and others train fully quantized models on Cortex-M MCUs (Deutel et al., 2024) or cut training memory with quantized parameter updates (Buron et al., 2025). These systems show that on-device backpropagation through quantized models is feasible, but they pay for it: they reintroduce the gradient machinery (an autograd graph, higher-precision master weights, optimizer state, and quantized-gradient bookkeeping) that a pure inference runtime omits, and they target on-device *training* rather than the inference-only deployment FORGE adapts within. FORGE occupies the opposite corner: it adds no gradient machinery and runs inside the deployed int8 inference engine, trading the generality of gradient updates for negligible cost.

**Quantization and on-device inference.** Integer-only inference (Jacob et al., 2018) folds batch normalization (Ioffe & Szegedy, 2015) into the preceding convolution and quantizes weights and activations to int8 (Krishnamoorthi, 2018; Nagel et al., 2021; Esser et al., 2020). This is standard for microcontroller runtimes (TFLite-Micro (David et al., 2021), CMSIS-NN (Lai et al., 2018), and ESP-NN (Espressif Systems, 2023)) and for tiny models such as MCUNet (Lin et al., 2020; 2021). The fold is precisely what removes the statistics FORGE restores.

**Positioning.** Table 1 summarizes the landscape against the properties a method needs to actually adapt a *deployed* MCU model. Each prior method satisfies a strict subset; ours is the only one that is forward-only, runs under true integer-only execution on a microcontroller, and reports measured energy.

## 3 Method

### 3.1 Folded integer-only deployment

For efficient MCU inference, each Conv→BN pair is fused into a single convolution with bias: $W' = W\gamma/\sqrt{\sigma^2 + \epsilon}$, $b' = \beta - \mu\gamma/\sqrt{\sigma^2 + \epsilon}$, and the BN layer is removed. This is exact at inference time, so clean accuracy is unchanged.

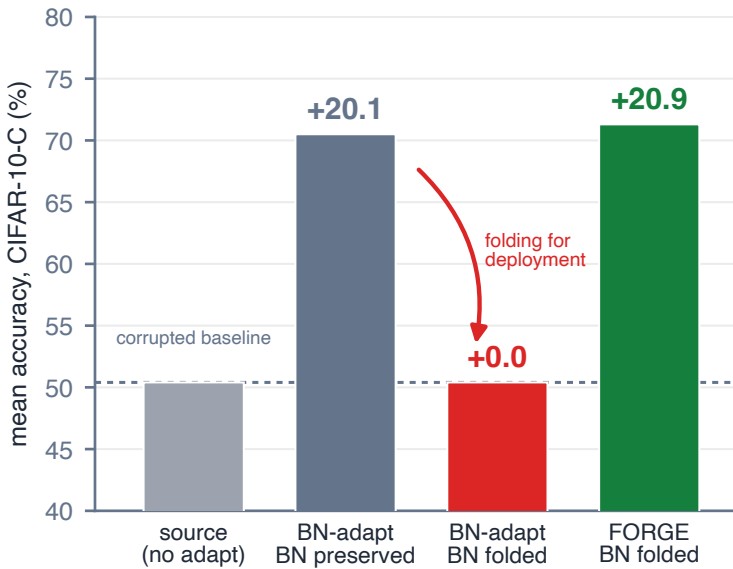

Figure 2: **The adaptation gap.** On CIFAR-10-C (int8, mean over the 15 corruptions), forward-only BN recalibration recovers +20.1 points on a BN-*preserving* model, but folding BN into the convolutions for integer deployment drops that to +0.0 (red): nothing remains to recalibrate, and the BN-preserving model is itself not deployable in integer-only form. FORGE restores +20.9 on the *same* deployed folded model (green; Sec. 4.1). The corruption error is identical across all bars.

### 3.2 The adaptation gap

Folding eliminates the running statistics $(\mu, \sigma^2)$ that BN-based adaptation updates. Empirically (Fig. 2), forward-only BN recalibration recovers +20.1 mean accuracy points on a BN-*preserving* int8 model, but +0.0 once BN is folded, since nothing remains to recalibrate, even though the corruption error is identical in both cases. The need for adaptation is unchanged. Only the remedy has vanished.

### 3.3 Forge: per-channel recalibration

We restore adaptation on the folded model by re-introducing a lightweight, gradient-free per-channel correction where each BN used to be.

**What folding removes, and what it leaves.** Before folding, a Conv→BN site normalizes its pre-activation to zero mean and unit variance using the running statistics $(\mu, \sigma^2)$, then applies the affine $(\gamma, \beta)$; under distribution shift, BN-adapt simply refreshes $(\mu, \sigma^2)$ from the test stream. Folding substitutes the *training-time* $(\mu, \sigma^2)$ into the convolution weights (Sec. 3.1) and deletes the layer, so $(\mu, \sigma^2)$ are no longer addressable: there is nothing for BN-adapt to refresh. What folding does *not* change is the clean per-channel output distribution of the fused convolution, which by construction still has mean $\beta_c$ and standard deviation $|\gamma_c|$. FORGE keeps exactly those two per-channel constants, recorded at fold time, as the recalibration target; no BN layer or stored feature bank is needed.

**Recalibration.** At test time we maintain an exponential moving average (EMA) of the per-channel output mean and variance over the (shifted) stream and re-normalize each channel back to its clean target (Alg. 1):

$$\hat{x}_c = \frac{x_c - \bar{\mu}_c}{\sqrt{\bar{\sigma}_c^2 + \epsilon}} \, |\gamma_c| + \beta_c,$$

where $(\bar{\mu}_c, \bar{\sigma}_c^2)$ are the running estimates, initialized to the clean target so an unshifted stream is a no-op. On a real channel of the deployed int8 model this snaps the corrupted output distribution back onto its clean

---

**Algorithm 1** FORGE recalibration at one former-BN site

---

**Require:** folded conv output $x \in \mathbb{R}^{B \times C \times H \times W}$; clean targets $(\beta_c, |\gamma_c|)$ from fold time; momentum $m$; running $(\bar{\mu}_c, \bar{\sigma}_c^2)$ (init. to $(\beta_c, \gamma_c^2)$)

1: $\mu_c \leftarrow \text{mean}_{B,H,W}(x_c); \quad v_c \leftarrow \text{var}_{B,H,W}(x_c)$         ▷ per-channel batch stats

2: $\bar{\mu}_c \leftarrow (1-m)\,\bar{\mu}_c + m\,\mu_c; \quad \bar{\sigma}_c^2 \leftarrow (1-m)\,\bar{\sigma}_c^2 + m\,v_c$         ▷ forward-only EMA

3: $\hat{x}_c \leftarrow \dfrac{x_c - \bar{\mu}_c}{\sqrt{\bar{\sigma}_c^2 + \epsilon}}\, |\gamma_c| + \beta_c$         ▷ re-center to clean target

4: **return** $\hat{x}_c$   (no gradients, no learnable parameters)

---

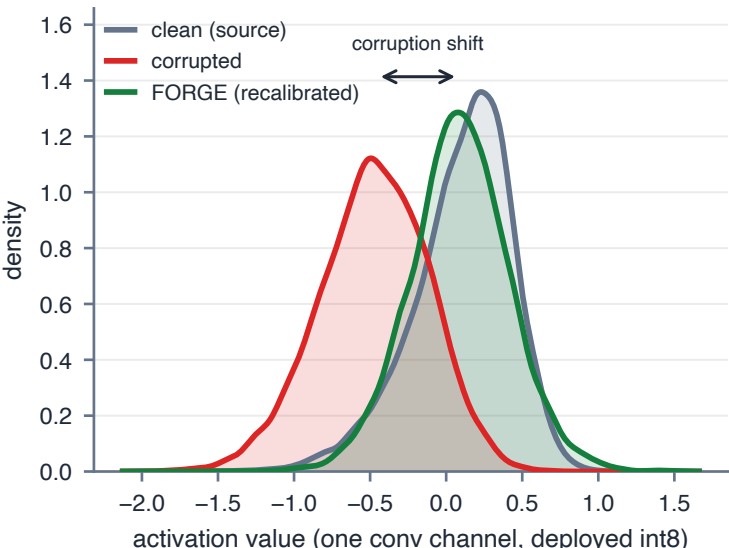

Figure 3: **The mechanism on real activations.** Measured per-channel output distribution of one convolution channel in the deployed int8 model (`layer2.0.bn1`, CIFAR-10-C Gaussian noise). Corruption displaces the channel by $1.8\sigma$ off its clean source mean (red); FORGE's forward-only recalibration restores it to within $0.06\sigma$ of the clean distribution (green), recovering both the mean and the spread. This is the per-channel correction of Alg. 1 measured on a real channel, not a schematic.

target (Fig. 3). This uses only forward passes and no learnable parameters, and unlike BN recalibration it requires no BN layers, only the $(\beta, |\gamma|)$ recorded at fold time. The cost is two reductions and an affine over the activation, gradient-free and carrying no optimizer state. Numerically, the running estimates $(\bar{\mu}_c, \bar{\sigma}_c^2)$ and the recalibration affine are computed in fp32, inside the requantization step that integer inference already performs on the MCU's FPU: the int8 convolution output is dequantized to fp32 (as it must be for the bias and output-scale requantization), Alg. 1 is applied there, and the result is re-quantized to int8 for the next layer. FORGE thus introduces no numeric regime beyond the fp32 requantization that int8 inference already runs, and no autograd graph, master weights, or optimizer state; "integer-only" here refers to the deployed inference path FORGE runs inside, not to integer arithmetic for the update itself. A naive alternative, adapting the activation quantization scales ("scale-adapt", a baseline we introduce rather than a prior method, hence absent from the Table 1 taxonomy), fails (Table 2): a per-tensor scale only stretches the dynamic range and cannot correct the per-channel mean/variance shift that corruption induces.

**Selective-layer recalibration.** Per-layer importance is highly non-uniform: a single layer's recovery ranges from $+0.4$ to $+17$ points. Adapting only the most important layers is both cheaper and, because the unhelpful layers slightly hurt, *more accurate* than adapting all layers (Sec. 4.2).

Table 2: Baselines on CIFAR-10-C (bs=64, full test set). The top group adapts the fp32 model (recovery vs the fp32 source); the bottom group adapts the deployed folded int8 model (recovery vs its int8 source, 50.4%). Only FORGE is forward-only, needs no BN layers, *and* runs on the folded integer-only MCU model. Subscripts are std over 5 random test-stream orderings. This table reports *measured accuracy*, so it also includes the unadapted source models and *scale-adapt* (a naive int8 baseline that adapts the per-tensor activation quantization scales, introduced here in Sec. 3); these are not adaptation methods from the literature and so do not appear in the Table 1 capability taxonomy, which lists only prior TTA methods and ours.

| method | acc. (%) | rec. | fwd | no BN | MCU |
|---|---|---|---|---|---|
| source (fp32) | 50.1 | — | ✓ | ✓ | ✓ |
| BN-adapt | 70.7 | $+20.6_{\pm 0.09}$ | ✓ | ✗ | ✗ |
| TENT (backprop) | 75.0 | $+24.9_{\pm 0.13}$ | ✗ | ✗ | ✗ |
| source (int8) | 50.4 | — | ✓ | ✓ | ✓ |
| scale-adapt | 50.0 | $-0.4$ | ✓ | ✓ | ✓ |
| **Forge** | **71.3** | $+20.9_{\pm 0.04}$ | ✓ | ✓ | ✓ |

**Single-sample streaming.** The deployment regime is batch size 1. Single-image statistics are noisy, so a fixed EMA momentum collapses; the effective averaging window is $\approx$ batch/momentum, so we scale momentum with batch size to hold the window constant (Sec. 4.3).

## 4 Experiments

**Setup.** ResNet-20 (He et al., 2016) and MobileNetV2 (Sandler et al., 2018; Howard et al., 2019) (0.5×) on CIFAR-10/100 (Krizhevsky, 2009) and Tiny-ImageNet (a 200-class, 64×64 ImageNet (Deng et al., 2009) subset); corruption benchmarks CIFAR-10/100-C and Tiny-ImageNet-C (Hendrycks & Dietterich, 2019) at severity 5. Models are folded and quantized to int8 (per-channel weights, per-tensor activations). We report mean accuracy / recovery over the 15 corruptions. Because online adaptation is order-dependent, recovery is reported as mean ± std over 5 random test-stream orderings (Tables 2, 4). On-device: ESP32-S3, energy via a Nordic PPK2. Some ablations use a fixed subset for compute; full-set runs are noted.

### 4.1 Comparison to baselines

FORGE matches forward-only BN adaptation and recovers most of backprop-based TENT's gain (+20.9 vs. +24.9), while being the only method that runs on the deployed folded integer-only model (Table 2). The ~4-point gap to TENT is the price of being gradient-free: TENT takes a gradient step per sample and keeps improving over the stream, whereas our running-statistic estimate converges and plateaus. This gap is, however, the wrong comparison for the deployment setting: neither TENT (it needs gradients) nor BN-adapt (it needs BN layers) can run on the folded integer-only model at all. On the device, the practical alternative to our +20.9 is not TENT but the unadapted model (+0.0; the folded source of Fig. 2). We also evaluate CoTTA (Wang et al., 2022) in its native continual protocol (the 15 corruptions as one stream): it recovers +13.9 points, but like TENT it relies on backpropagation (and augmentation forward passes), so it too cannot run on the folded integer-only model (Table 1).

**Per-corruption behaviour.** Table 3 breaks the comparison out over all 15 corruptions. FORGE tracks forward-only BN-adapt closely on every corruption while running on the folded int8 model, and the gains are largest exactly where they matter most: severe corruptions that collapse the source model (contrast $25.4 \rightarrow 79.0$, Gaussian noise $25.6 \rightarrow 60.3$). On already-mild corruptions where the source is strong (brightness, JPEG) it is flat to slightly negative, the standard behaviour of normalization-based adaptation, which the safety gate of Sec. 4.6 removes.

Table 3: Per-corruption accuracy (%) on CIFAR-10-C (severity 5, bs=64). "source" is the fp32 model; FORGE runs on the folded int8 model. FORGE matches forward-only BN-adapt across the board and recovers most on the hardest corruptions.

| corruption | source | BN-adapt | TENT | FORGE |
|---|---|---|---|---|
| Gaussian noise | 25.6 | 59.7 | 67.3 | 60.3 |
| Shot noise | 31.1 | 60.6 | 69.3 | 61.4 |
| Impulse noise | 30.8 | 54.0 | 61.1 | 54.3 |
| Defocus blur | 45.0 | 81.4 | 83.2 | 82.0 |
| Glass blur | 40.6 | 56.5 | 60.5 | 56.6 |
| Motion blur | 56.1 | 78.7 | 81.1 | 79.4 |
| Zoom blur | 49.5 | 80.3 | 83.2 | 80.6 |
| Snow | 67.5 | 71.7 | 76.8 | 72.4 |
| Frost | 52.1 | 71.7 | 76.3 | 72.4 |
| Fog | 63.4 | 77.4 | 80.4 | 78.1 |
| Brightness | 86.8 | 84.8 | 87.1 | 85.7 |
| Contrast | 25.4 | 78.6 | 79.2 | 79.0 |
| Elastic | 67.4 | 68.2 | 71.7 | 68.9 |
| Pixelate | 41.4 | 70.2 | 76.7 | 70.7 |
| JPEG | 69.0 | 66.2 | 71.3 | 67.2 |
| mean | 50.1 | 70.7 | 75.0 | 71.3 |

## 4.2 Selective-layer recalibration

We stress that layer selection is a *one-time, design-time* step performed by the developer *before* deployment, using only source data and held-out synthetic corruptions: it never touches the test stream and never runs on the device, so it does not require test-time data and does not conflict with the deployed-model, no-data premise of TTA. It is also *optional*: adapting *all* 21 layers needs no selection or calibration set whatsoever and already recovers +18.4 (below); selection is a cheap offline optimization on top, not a prerequisite. Concretely, ranking layer importance on a held-out set of corruptions and evaluating on a disjoint set, five of the six most important layers are shared across the split (importance is largely corruption-independent, so the ranking transfers without ever seeing the test corruptions). Adapting the top-3 layers (chosen on held-out corruptions) recovers 93% of the full benefit on unseen corruptions, matching an oracle that sees the test set (Fig. 4). Adapting the best subset *exceeds* adapting all 21 layers (the held-out selection peaks at +19.6 with 8 layers versus +18.4 for all 21), because the unhelpful layers slightly hurt.

## 4.3 Single-sample streaming

A fixed momentum holds down to batch 4 but collapses at batch 1; window-matched momentum ($m = \text{bs}/640$) holds (Fig. 5).

## 4.4 Generalization

FORGE generalizes across datasets and architectures (Table 4): ResNet-20 and MobileNetV2 (a depthwise-separable, inverted-residual family) recover nearly identically on CIFAR-10-C, and the method holds on the harder 100-class CIFAR-100-C. The MobileNetV2 result matters beyond a second data point: the recalibration does *not* rely on standard ("vanilla") convolutions—it applies unchanged to the depthwise and pointwise layers that make up the efficient, separable backbones used on edge devices, since every such layer is still a folded `Conv→BN` site with a per-channel output distribution to restore. It also scales to Tiny-ImageNet-C (200 classes, 64×64; corruptions generated with the `imagecorruptions` library (Michaelis et al., 2019)), where it recovers +11.0 points with *all* 15 corruptions improving. This is a smaller absolute gain, as expected on a much harder benchmark (53% clean) where severity-5 corruptions are devastating, but it is a clear positive across the board.

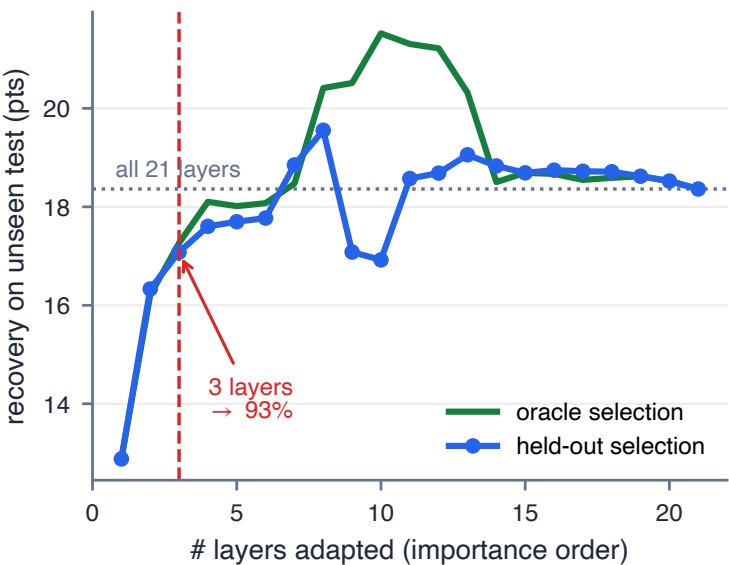

Figure 4: Held-out selective recalibration (full CIFAR-10-C test). Layers ranked on held-out corruptions; recovery shown on *unseen* test corruptions. A few layers reach 93% of full recovery (knee at 3), held-out tracks the oracle, and the best subset exceeds adapting all 21 layers.

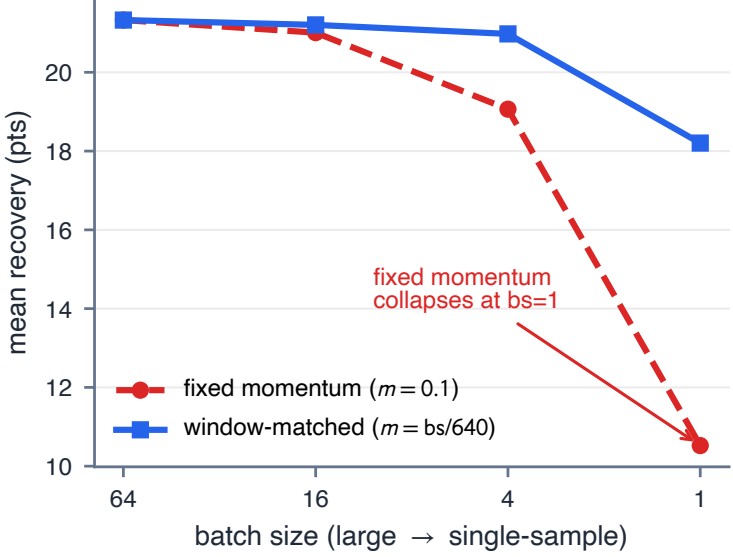

Figure 5: Recovery vs. batch size (CIFAR-10-C). A fixed EMA momentum collapses at single-sample streaming; scaling momentum with batch size (constant averaging window $\approx$ bs/$m$) holds.

### 4.5 Ablations

**Momentum.** FORGE has a single hyperparameter, the EMA momentum. Recovery is remarkably flat in it (Table 5): sweeping $m$ across two orders of magnitude (0.01 to 1.0) moves mean recovery by under 0.8 points, peaking at $m$=0.1. The method therefore needs no per-deployment tuning of its only knob.

Table 4: Generalization: mean recovery (bs=64). Two architectures × three datasets (CIFAR-10-C, CIFAR-100-C, and Tiny-ImageNet-C / TIN-C: 200 classes, 64×64). Subscripts are std over 5 test-stream orderings; TIN-C is single-seed.

|  | CIFAR-10-C | CIFAR-100-C | TIN-C |
|---|---|---|---|
| ResNet-20 | $+20.9_{\pm 0.04}$ | $+14.5_{\pm 0.06}$ | — |
| MobileNetV2 | $+20.5_{\pm 0.09}$ | — | $+11.0$ |

Table 5: Momentum ablation (ResNet-20, CIFAR-10-C, int8, bs=64). Mean recovery vs. the int8 source is stable across two orders of magnitude of the EMA momentum.

| momentum $m$ | 0.01 | 0.03 | 0.1 | 0.3 | 1.0 |
|---|---|---|---|---|---|
| recovery | $+20.2$ | $+20.4$ | $+20.8$ | $+20.7$ | $+20.1$ |

Table 6: Bit-width ablation (ResNet-20, CIFAR-10-C). FORGE recovers across precisions; the magnitude tracks the headroom (lower source = more to restore) and the mechanism holds even at 4-bit weights and activations.

| (weight, act.) bits | 8/8 | 6/6 | 4/8 | 4/4 |
|---|---|---|---|---|
| clean acc. (%) | 91.9 | 90.5 | 90.1 | 60.6 |
| source acc. (%) | 50.4 | 50.2 | 47.6 | 32.2 |
| recovery | $+20.8$ | $+18.5$ | $+23.3$ | $+13.3$ |

**Bit width.** Recalibration is not specific to int8. As weight/activation precision drops (Table 6), FORGE keeps recovering: at aggressive 4-bit weights it recovers $+23.3$ (the quantized source is lower, so there is more to restore), and even at 4/4—where quantization alone collapses clean accuracy to 60.6%—it still recovers $+13.3$. The recovery magnitude tracks the available headroom, but the mechanism itself never breaks, which is what we would expect of a correction that re-centers distributions rather than relying on a particular numeric range.

### 4.6 When to adapt: a safety gate

Normalization-based adaptation can slightly hurt on already-mild corruptions where the deployed model is fine (Table 3: brightness, JPEG). A forward-only gate removes this downside. The naive signal, absolute prediction confidence, does not separate the cases: JPEG (mean top-1 confidence 0.855, adaptation hurts) and snow (0.857, adaptation helps $+4.8$) are indistinguishable, so a confidence threshold either fails to protect JPEG or needlessly drops snow. The signal that does work is whether adaptation *improves* the model's own confidence. We keep recalibration on a stream only if it raises mean top-1 confidence over the unadapted model, a self-supervised check that needs no labels and only the forward passes the model already runs. This keeps adaptation on the 13 corruptions where it helps and reverts to the source on brightness and JPEG, turning the 2 degraded corruptions into 0 while slightly *raising* mean recovery ($+20.8 \rightarrow +21.0$; Table 7). Adaptation becomes strictly safe at the cost of one confidence comparison during a short calibration window.

### 4.7 On-device deployment

We export the folded int8 model and verify a true integer-only reference reproduces the PyTorch path (clean 92.6% vs. 92.2%). On the ESP32-S3 the C inference matches the reference logits exactly (max $|\Delta| = 0.0$), so the deployed model is the same one the simulation results describe. Inference uses the ESP-NN SIMD-optimized int8 kernels, and the recalibration itself *runs on the physical ESP32-S3, not in a simulator*: both the inference and the inference+adapt numbers below are measured on the same on-device firmware whose logits match the integer reference exactly. Energy is measured with a Nordic PPK2 in source mode (3.3 V), integrating current at 100 kHz; inference and inference+adapt are separated by their distinct durations.

Table 7: When-to-adapt gate (CIFAR-10-C, ResNet-20, int8; single representative run, so "always adapt" is within seed noise of Table 2). Keeping recalibration only when it raises the deployed model's own top-1 confidence makes adaptation safe—no corruption is degraded—and leaves mean recovery unchanged-to-better. Absolute confidence cannot separate the helped from the hurt corruptions.

| policy | mean recovery | corruptions hurt |
|---|---|---|
| always adapt | +20.8 | 2 / 15 |
| gate: absolute conf. | +20.9 | 1 / 15 |
| gate: conf. improvement | **+21.0** | **0 / 15** |

Inference runs on the optimized ESP-NN SIMD int8 kernels at $286\,\mathrm{ms}$ ($9.6\times$ faster than a naive convolution), and adaptation adds only $8.3\,\mathrm{mJ}$ (6.8% of inference energy) and $21.9\,\mathrm{ms}$ (Table 8): forward-only adaptation is cheap on a microcontroller. The reason is structural. FORGE adds, per former-BN site, two per-channel reductions (mean and variance) and one per-channel affine over the activation; there is no weight update, no second forward pass, and no gradient buffer, so the overhead is a *fixed* $8.3\,\mathrm{mJ}$ regardless of how fast the convolutions run, scaling with the *activation* volume the network already streams through rather than the parameter count. Memory is equally modest: the only state FORGE adds is a per-channel running mean and variance at each former-BN site—784 values across the 21 sites of ResNet-20, i.e. $\approx 6\,\mathrm{KB}$ in fp32 (and $\approx 1$–$2\,\mathrm{KB}$ for the selective top-3 of Sec. 3.3). Recalibration is applied *layer-by-layer and in place*: each site corrects its own activation as the forward pass reaches it, reusing the activation buffer inference already holds, so it adds no extra activation-sized buffer and does not raise the network's peak memory footprint. The headline contrast is not with TENT's accuracy but with its deployability: backpropagation needs the gradient graph and optimizer state that an integer-only MCU runtime cannot hold, whereas FORGE runs in the same forward, integer-only path as inference. No prior forward-only TTA method reports this measured on-device cost.

Table 8: Measured on-device energy (ESP32-S3, PPK2, 3.3 V) with the deployed ESP-NN SIMD int8 kernels. The adaptation overhead is a fixed cost, independent of the inference kernel.

| operation | energy (mJ) | time (ms) | power (mW) |
|---|---|---|---|
| inference | 121.7 | 286.1 | 425 |
| inference + adapt | 130.0 | 308.0 | 422 |
| **adapt overhead** | **8.3** | **21.9** | — |

## 5 Limitations

The recalibration mechanism is close to BN-statistic adaptation; our contribution is making it work on a *deployed* folded integer-only model and measuring its on-device cost, not a new adaptation principle. The confidence gate of Sec. 4.6 makes adaptation safe within CIFAR-10-C, but we have not validated the gate threshold across datasets, and on-device validation is on ResNet-20 while the second architecture is evaluated in simulation. FORGE also targets the BN-fold regime specifically: it assumes folded Conv→BN sites, so it covers the convolutional backbones common on microcontrollers (including the depthwise-separable family above) but not transformer backbones, whose LayerNorm is not fused into a preceding convolution in the same way. Extending the same post-fold restoration idea to other normalization layers is natural future work.

## 6 Conclusion

Deploying a vision model on a microcontroller, which folds BN and quantizes to integers, silently removes its ability to adapt to the distribution shift it will face in the field. We restore that ability with a forward-only per-channel recalibration that runs on the deployed integer-only model, matches gradient-based adaptation

in accuracy, needs only a few layers, survives single-sample streaming, generalizes across datasets and architectures, and, measured on real hardware, costs 6.8% of inference energy. Forward-only test-time adaptation is practical, and essentially free, on a microcontroller.

## A  Why recalibration cancels the corruption, and the variance it omits

**Setup.**  At a folded `Conv→BN` site, the clean output of channel $c$ is $y_c = \gamma_c z_c + \beta_c$, where $z_c = (u_c - \mu_c)/\sqrt{\sigma_c^2 + \epsilon}$ is the batch-normalized pre-activation. By construction $z_c$ has zero mean and unit variance on the training distribution, so $\mathbb{E}[y_c] = \beta_c$ and $\mathrm{Std}[y_c] = |\gamma_c|$: these are the two per-channel constants FORGE records at fold time, and the only quantities it retains about the clean model.

**Exact cancellation under an affine channel shift.**  To first order, a covariate shift propagates through one folded linear+ReLU site as a per-channel affine map of the clean output, $\tilde{y}_c = a_c\, y_c + d_c$ with scale $a_c > 0$ and offset $d_c$—exactly the "$x = A\, x_{\mathrm{orig}} + B$" regime. Its moments are $\mathbb{E}[\tilde{y}_c] = a_c\beta_c + d_c$ and $\mathrm{Var}[\tilde{y}_c] = a_c^2\gamma_c^2$. FORGE's running estimates converge to these, $\bar{\mu}_c \to \mathbb{E}[\tilde{y}_c]$ and $\bar{\sigma}_c^2 \to \mathrm{Var}[\tilde{y}_c]$, and the recalibration (Alg. 1) gives, as $\epsilon \to 0$,

$$\hat{x}_c = \frac{\tilde{y}_c - \bar{\mu}_c}{\sqrt{\bar{\sigma}_c^2 + \epsilon}}\, |\gamma_c| + \beta_c = \frac{a_c y_c + d_c - (a_c\beta_c + d_c)}{\sqrt{a_c^2\gamma_c^2}}\, |\gamma_c| + \beta_c = \frac{a_c\, (y_c - \beta_c)}{a_c|\gamma_c|}\, |\gamma_c| + \beta_c = y_c.$$

The clean output is recovered *exactly*, and the corruption coefficients cancel: the standardization divides out the scale $a_c$ independently of its value, and subtracting $\bar{\mu}_c$ removes the offset $d_c$. Tracking $(\bar{\mu}_c, \bar{\sigma}_c^2)$ and rescaling thus acts as the inverse of the affine corruption map, dropping the multiplicative coefficient—no property of the corruption need be known, only the fold-time $(\beta_c, |\gamma_c|)$.

**The variance the EMA omits (law of total variance).**  The cancellation above uses the *true* $\mathrm{Var}[\tilde{y}_c]$. FORGE estimates it with an EMA of per-batch variances $v_i = \mathrm{Var}(\tilde{y}_c \mid \text{batch } i)$, which for a stationary stream converges to $\mathbb{E}[\mathrm{Var}(\tilde{y}_c \mid \text{Batch})]$. By the law of total variance,

$$\mathrm{Var}(\tilde{y}_c) = \underbrace{\mathbb{E}[\mathrm{Var}(\tilde{y}_c \mid \text{Batch})]}_{\text{tracked by the EMA}} + \underbrace{\mathrm{Var}(\mathbb{E}[\tilde{y}_c \mid \text{Batch}])}_{\text{omitted}},$$

so the estimate falls short by the between-batch-mean term, and the recalibration divides by a slightly *under-estimated* standard deviation. If the omitted term is a fraction $f$ of the total variance, the residual scaling error is $1/\sqrt{1-f} - 1 \approx f/2$. For random batches of size $B$ from a stream whose per-image channel means have variance $\sigma_\mu^2$, the omitted term is exactly $\mathrm{Var}(\mathbb{E}[\tilde{y}_c \mid \text{Batch}]) = (\sigma_\mu^2/B)\,(N - B)/(N - 1)$: it shrinks as $1/B$.

We measure $f$ directly on the deployed int8 model, over all 783 channels of the 21 recalibration sites (Table 9). At the batch size of our main results ($B{=}64$) the omitted term is $\approx 0.1\%$ of the total variance—a $\approx 0.05\%$ scaling error, far below the int8 quantization step itself—so the cancellation holds to high precision. As $B$ falls the term grows as $1/B$, reaching $\approx 8\%$ (a $\approx 4.7\%$ scaling error) at single-sample streaming on Gaussian noise, and more for the most severe shifts (16% for contrast, 10% for fog). This is precisely why a fixed-momentum estimator degrades at $B{=}1$ (Sec. 4.3, Fig. 5): the window-matched momentum $m = \text{bs}/640$ we adopt holds the effective averaging window—and hence this between-batch term—constant as the batch shrinks, which is the practical control for the bias the decomposition identifies.

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

Table 9: The between-batch variance term the EMA omits, measured on the deployed int8 model (ResNet-20, CIFAR-10-C Gaussian noise, severity 5; mean over all 783 channels of the 21 recalibration sites). The omitted term is a fraction of the true total variance and shrinks as $1/B$; the implied error in the 1/std rescale is $\approx f/2$. It is negligible at the $B{=}64$ of the main results and dominates only at single-sample streaming, where the window-matched momentum of Sec. 4.3 controls it.

| batch size $B$ | 1 | 2 | 4 | 8 | 16 | 32 | 64 |
|---|---|---|---|---|---|---|---|
| omitted $\mathrm{Var}(\mathbb{E}[X|B])/\mathrm{Var}(X)$ | 7.96% | 3.97% | 1.98% | 0.98% | 0.48% | 0.23% | 0.11% |
| implied 1/std scaling error | 4.73% | 2.14% | 1.03% | 0.50% | 0.24% | 0.12% | 0.05% |

Zeshuai Deng, Guohao Chen, Shuaicheng Niu, Hui Luo, Shuhai Zhang, Yifan Yang, Renjie Chen, Wei Luo, and Mingkui Tan. Test-time model adaptation for quantized neural networks. *arXiv:2508.02180*, 2025.

Mark Deutel, Frank Hannig, Christopher Mutschler, and Jürgen Teich. On-device training of fully quantized deep neural networks on cortex-M microcontrollers. *IEEE Transactions on Computer-Aided Design of Integrated Circuits and Systems*, 44(4):1250–1261, 2024.

Cynthia Dong, Hong Jia, Young D. Kwon, Georgios Rizos, and Cecilia Mascolo. Leantta: A backpropagation-free and stateless approach to quantized test-time adaptation on edge devices. *arXiv:2503.15889*, 2025.

Espressif Systems. Esp-nn: Optimized nn functions for espressif socs. `https://github.com/espressif/esp-nn`, 2023.

Steven K. Esser, Jeffrey L. McKinstry, Deepika Bablani, Rathinakumar Appuswamy, and Dharmendra S. Modha. Learned step size quantization. In *ICLR*, 2020.

Taesik Gong, Jongheon Jeong, Taewon Kim, Yewon Kim, Jinwoo Shin, and Sung-Ju Lee. Note: Robust continual test-time adaptation against temporal correlation. In *NeurIPS*, 2022.

Kaiming He, Xiangyu Zhang, Shaoqing Ren, and Jian Sun. Deep residual learning for image recognition. In *CVPR*, 2016.

Xiangyu He and Jian Cheng. Learning compression from limited unlabeled data. In *ECCV*, 2018.

Dan Hendrycks and Thomas Dietterich. Benchmarking neural network robustness to common corruptions and perturbations. In *ICLR*, 2019.

Andrew Howard, Mark Sandler, Grace Chu, Liang-Chieh Chen, Bo Chen, Mingxing Tan, et al. Searching for mobilenetv3. In *ICCV*, 2019.

Sergey Ioffe and Christian Szegedy. Batch normalization: Accelerating deep network training by reducing internal covariate shift. In *ICML*, 2015.

Yusuke Iwasawa and Yutaka Matsuo. Test-time classifier adjustment module for model-agnostic domain generalization. In *NeurIPS*, 2021.

Benoit Jacob, Skirmantas Kligys, Bo Chen, Menglong Zhu, Matthew Tang, Andrew Howard, Hartwig Adam, and Dmitry Kalenichenko. Quantization and training of neural networks for efficient integer-arithmetic-only inference. In *CVPR*, 2018.

Hong Jia, Young D. Kwon, Alessio Orsino, Ting Dang, Domenico Talia, and Cecilia Mascolo. Tinytta: Efficient test-time adaptation via early-exit ensembles on edge devices. In *NeurIPS*, 2024.

Raghuraman Krishnamoorthi. Quantizing deep convolutional networks for efficient inference: A whitepaper. *arXiv:1806.08342*, 2018.

Alex Krizhevsky. Learning multiple layers of features from tiny images. Technical report, University of Toronto, 2009.

Liangzhen Lai, Naveen Suda, and Vikas Chandra. Cmsis-nn: Efficient neural network kernels for arm cortex-m cpus. *arXiv:1801.06601*, 2018.

Yanghao Li, Naiyan Wang, Jianping Shi, Xiaodi Hou, and Jiaying Liu. Adaptive batch normalization for practical domain adaptation. *Pattern Recognition*, 80:109–117, 2018.

Jian Liang, Dapeng Hu, and Jiashi Feng. Do we really need to access the source data? source hypothesis transfer for unsupervised domain adaptation. In *ICML*, 2020.

Ji Lin, Wei-Ming Chen, Yujun Lin, John Cohn, Chuang Gan, and Song Han. Mcunet: Tiny deep learning on iot devices. In *NeurIPS*, 2020.

Ji Lin, Wei-Ming Chen, Han Cai, Chuang Gan, and Song Han. Mcunetv2: Memory-efficient patch-based inference for tiny deep learning. In *NeurIPS*, 2021.

Ji Lin, Ligeng Zhu, Wei-Ming Chen, Wei-Chen Wang, Chuang Gan, and Song Han. On-device training under 256KB memory. In *Advances in Neural Information Processing Systems (NeurIPS)*, volume 35, pp. 22941–22954, 2022.

Xiao Ma, Young D. Kwon, Pan Zhou, and Dong Ma. Architecture-agnostic test-time adaptation via backprop-free embedding alignment. In *ICLR*, 2026.

Claudio Michaelis, Benjamin Mitzkus, Robert Geirhos, Evgenia Rusak, Oliver Bringmann, Alexander S. Ecker, Matthias Bethge, and Wieland Brendel. Benchmarking robustness in object detection: Autonomous driving when winter is coming. *arXiv:1907.07484*, 2019. imagecorruptions library.

Zachary Nado, Shreyas Padhy, D. Sculley, Alexander D'Amour, Balaji Lakshminarayanan, and Jasper Snoek. Evaluating prediction-time batch normalization for robustness under covariate shift. *arXiv:2006.10963*, 2020.

Markus Nagel, Marios Fournarakis, Rana Ali Amjad, Yelysei Bondarenko, Mart van Baalen, and Tijmen Blankevoort. A white paper on neural network quantization. *arXiv:2106.08295*, 2021.

Shuaicheng Niu, Jiaxiang Wu, Yifan Zhang, Yaofo Chen, Shijian Zheng, Peilin Zhao, and Mingkui Tan. Efficient test-time model adaptation without forgetting. In *ICML*, 2022.

Shuaicheng Niu, Jiaxiang Wu, Yifan Zhang, Zhiquan Wen, Yaofo Chen, Peilin Zhao, and Mingkui Tan. Towards stable test-time adaptation in dynamic wild world. In *ICLR*, 2023.

Shuaicheng Niu, Chunyan Miao, Guohao Chen, Pengcheng Wu, and Peilin Zhao. Test-time model adaptation with only forward passes. In *ICML*, 2024. arXiv:2404.01650.

Mark Sandler, Andrew Howard, Menglong Zhu, Andrey Zhmoginov, and Liang-Chieh Chen. Mobilenetv2: Inverted residuals and linear bottlenecks. In *CVPR*, 2018.

Steffen Schneider, Evgenia Rusak, Luisa Eck, Oliver Bringmann, Wieland Brendel, and Matthias Bethge. Improving robustness against common corruptions by covariate shift adaptation. In *NeurIPS*, 2020.

Damian Sójka, Sebastian Cygert, and Marc Masana. Subspace optimization for backpropagation-free continual test-time adaptation. *arXiv:2603.28678*, 2026.

Dequan Wang, Evan Shelhamer, Shaoteng Liu, Bruno Olshausen, and Trevor Darrell. Tent: Fully test-time adaptation by entropy minimization. In *ICLR*, 2021.

Qin Wang, Olga Fink, Luc Van Gool, and Dengxin Dai. Continual test-time domain adaptation. In *CVPR*, 2022.

Marvin Zhang, Sergey Levine, and Chelsea Finn. Memo: Test-time robustness via adaptation and augmentation. In *NeurIPS*, 2022.

