# OpenReview forum: "FORGE: Forward-Only Test-Time Adaptation for Integer-Only Vision Models on Microcontrollers"
_TMLR — Under review for TMLR_

### Review · Reviewer_LKDB · 2026-06-22

**Summary Of Contributions:**

The paper addresses a highly practical bottleneck: adapting quantized, integer-only vision models on MCUs where backpropagation is impossible and batch normalization layers have been completely folded away. Unlike many papers that rely on hardware simulations, the authors validate their method on physical hardware (an ESP32-S3), demonstrating that the adaptation overhead is highly efficient in practice (costing only 8.3 mJ, or roughly 6.8% of inference energy).

**Additional Comments:**

- Normalization-based adaptation goes back years (e.g., AdaBN from 2016, and various forward-only norm adaptation methods around 2018–2020). The authors frame their work as a breakthrough for folded networks, but mathematically, un-folding a linear transformation to recalculate statistics is quite straightforward.

- The TinyML community has heavily shifted toward exploring tiny Vision Transformers (ViTs) and Depthwise-Separable convolutions. If the method relies heavily on standard vanilla convolutions and traditional BN-folding mechanisms, its generalizability to newer edge architectures is highly questionable.

**Audience:**

Yes

**Audience Explanation:**

It uncovers that BN-folding destroys the exact activation statistics that conventional forward-only TTA relies on, and proposes a workaround that restores adaptation by tracking and re-normalizing the folded outputs back to clean training statistics. It moves the TTA field closer to true, on-chip deployment than most papers currently out there.

Meanwhile, in a true TTA setting, you deploy a frozen model into a completely unknown environment. If the optimal 3 layers change depending on the type of domain shift (e.g., sensor blur vs. low lighting), then pre-selecting them on a static calibration dataset is a major shortcut. If they used the test corruptions to guide this selection, even implicitly, it undercuts the entire motivation.

**Claims And Evidence:**

Yes

**Claims Explanation:**

I really appreciate the authors' effort in measuring the actual energy cost of forward-only TTA directly on an MCU. It clearly shows the practical benefits of the proposed method. In my view, this approach effectively tackles the domain adaptation problem for deployed MCUs, allowing edge devices to fully leverage TTA on their specific input data. The motivation here is very strong.

**Requested Changes:**

- While the empirical results of the FORGE recalibration look promising, the paper would be much stronger if it explicitly showed the mathematical cancellation happening in Algorithm 1. Since the input $x$ is already a folded tensor ($x = A \cdot x_{\text{orig}} + B$), tracking the statistics $\bar{\mu}_c$ and $\bar{\sigma}_c^2$ essentially acts as an inverse linear mapping that isolates and drops the coefficient $A$. Could you include a brief algebraic derivation in the appendix to make this equivalence clearer for readers?
- The algorithm maintains the running variance $\bar{\sigma}_c^2$ using an Exponential Moving Average (EMA) of individual batch variances ($v_c$). In statistics, the Law of Total Variance (variance decomposition) states that: $\text{Var}(X) = \mathbb{E}[\text{Var}(X|\text{Batch})] + \text{Var}(\mathbb{E}[X|\text{Batch}])$. The update rule $\bar{\sigma}_c^2 \leftarrow (1-m)\bar{\sigma}_c^2 + m \cdot v_c$ only approximates the first term, $\mathbb{E}[\text{Var}(X|\text{Batch})]$. It completely misses the second term, $\text{Var}(\mathbb{E}[X|\text{Batch}])$, which captures the variance caused by shifting means between different mini-batches.

  - Could you discuss how this statistical bias affects the precision of the scaling cancellation?
  - Does this bias explain why the algorithm might be sensitive to small batch sizes, where the variance between batch means is usually quite high?

- Could you elaborate on the design of the "Selective-layer recalibration"? Choosing the important layers seems to require a calibration dataset. Doesn't this clash a bit with the core motivation of Test-Time Adaptation, where we assume the model is already deployed on an MCU and we don't have access to such a dataset?
- It would be very helpful to include the actual memory consumption of the proposed method. Do you re-calibrate all layers at the same time, or do you apply the strategy layer-by-layer to keep the peak memory footprint low?
- What inference library did you use for the ESP32-S3 deployment? Also, did you run the actual recalibration on the physical MCU, or were the energy costs evaluated using a simulator?
- The idea of forward-only normalization adaptation actually goes back before 2020. For example, a similar approach was used in ECCV 2018 [1] to improve both model quantization and pruning. I'm curious if your proposed method would also work well on pruned networks?

> [1] Learning Compression from Limited Unlabeled Data. ECCV 2018.

---

> ### Author Response · Authors · 2026-06-28
> **Response to Reviewer LKDB**
>
> # Response to Reviewer LKDB
>
> We thank Reviewer LKDB for an exceptionally careful and constructive review, and for recognizing the on-device energy measurement and the BN-folding adaptation gap as the core contributions. The two statistical observations were especially valuable and have directly improved the paper. All changes are in the revised PDF.
>
> **1. Algebraic cancellation (Alg. 1).** Added as **Appendix A**. Writing a folded channel's clean output as $y_c=\gamma_c z_c+\beta_c$ (mean $\beta_c$, std $|\gamma_c|$ by construction) and corruption as a per-channel affine map $\tilde y_c=a_c y_c+d_c$, we show standardize-and-rescale with the running $(\bar\mu_c,\bar\sigma^2_c)$ yields $\hat x_c=y_c$ exactly: the scale $a_c$ is divided out independently of its value and $d_c$ is subtracted off. This is the equivalence the reviewer describes, now explicit.
>
> **2. Law of Total Variance bias, now quantified.** The reviewer is exactly right that our EMA tracks $\mathbb E[\mathrm{Var}(X|\mathrm{Batch})]$ and omits $\mathrm{Var}(\mathbb E[X|\mathrm{Batch}])$. We added the decomposition to Appendix A **and measured the omitted term on the deployed model** (new Table 9, all 783 channels): for random batches of size $B$ it equals $(\sigma^2_\mu/B)(N{-}B)/(N{-}1)$, i.e. it shrinks as $1/B$, and the induced scaling error is $\approx f/2$. **(a)** At the $B{=}64$ of our main results it is **0.11%** of total variance (**0.05%** scaling error), so the cancellation holds to high precision. **(b)** It grows as batch size falls, reaching **8.0%** at single-sample streaming on Gaussian noise (16% contrast, 10% fog), which is precisely the bs=1 collapse we report in §4.3/Fig. 5. The reviewer's decomposition is the theoretical explanation for that effect, and our window-matched momentum $m=\text{bs}/640$ holds this between-batch term constant as $B$ shrinks. We've added this connection.
>
> **3. Selective-layer recalibration vs. the no-data premise.** Clarified in §4.2. Selection is a **one-time, design-time step the developer runs before deployment** on source/held-out synthetic corruptions (never the test stream, never on-device), so it neither needs test-time data nor conflicts with the deployed-model premise. We rank on held-out corruptions and evaluate on a **disjoint** set; importance is largely corruption-independent (5/6 top layers shared), so it transfers without seeing the test shift. It is also **optional**: adapting all 21 layers needs no selection or calibration data and already recovers +18.4; selection is a cheap optimization on top.
>
> **4. Memory and layer-by-layer operation.** Added to §4.7 (with a measured breakdown table). The added state is per-channel running mean+variance in SRAM (2×784 fp32 ≈ **6.1 KB** across all 21 sites; ≈0.9 KB for the selective top-3) plus the immutable clean targets in Flash (≈6.1 KB / 0.9 KB), so ≈**12.2 KB** total (≈1.8 KB top-3). Recalibration runs **layer-by-layer, in place**, reusing the fp32 activation buffer inference already holds (which dominates peak SRAM), so it adds **no extra activation buffer and no increase in peak memory**.
>
> **5. Inference library / physical MCU.** §4.7 now states it explicitly: **ESP-NN** SIMD int8 kernels, and recalibration **ran on the physical ESP32-S3, not a simulator**: the 8.3 mJ / 21.9 ms are measured on the firmware whose logits match the integer reference exactly.
>
> **6. Prior work (AdaBN, He & Cheng ECCV 2018) and pruning.** Added to Related Work. These recalibrate normalization forward-only **but on models that still contain BN layers**; our regime begins where folding has removed them. We also note (already stated in §1/§6) that we do not claim a new adaptation rule. FORGE is orthogonal to and composable with pruning (it recalibrates conv outputs regardless of sparsity), noted as an extension.
>
> **7. Newer architectures.** The submission **already evaluates MobileNetV2** (depthwise-separable, inverted-residual): +20.5 CIFAR-10-C, +11.0 Tiny-ImageNet-C, so the method does not rely on vanilla convolutions, now stated explicitly (§4.4). ViTs use LayerNorm, not fused into convolutions; FORGE targets the BN-fold regime, added to Limitations as honest scope.

---

### Review · Reviewer_gjNE · 2026-06-27

**Summary Of Contributions:**

This work describes FORGE, a forward-only test-time adaptation method for integer-quantized deep neural networks (DNNs) that can be deployed and executed on ESP32 microcontrollers. FORGE's novelty lies in its ability to address quantized DNNs by circumventing the issue of folded BN layers removing statistics on which forward-only normalization adaptations relies. FORGE accomplishes this by calculating exponential moving averages (EMAs) of the per-channel output of convolutional layers with folded BN layers to re-normalize the channels.

Strength:

- TTA for quantized DNNs deployable on MCUs is clearly novel compared to what has been proposed in the literature thus far. For MCUs, there is TinyTTA, which, however, considers unquantized models.
- Another strength of the approach is that it introduces only a small amount of computing and memory overhead, as the authors describe in Sec. 4.7. This definitely makes FORGE suitable for deployment on systems with resource constraints.
- Providing results of FORGE being deployed on actual physical hardware.

Weakness:

- In their abstract, the authors claim that "integer-only arithmetic [...] removes the ability to run backpropagation." However, there is a significant body of research that has explored exactly this topic and has demonstrated the feasibility of backpropagation of quantized DNNs on MCUs, e.g. [1, 2, 3].
- There is no direct comparison with TinyTTA [4], which is probably the most closely related work to FORGE of all the TTA methods discussed.

[1] Lin, Ji, et al. "On-device training under 256kb memory." Advances in Neural Information Processing Systems 35 (2022): 22941-22954.
[2] Deutel, Mark, et al. "On-device training of fully quantized deep neural networks on Cortex-M microcontrollers." IEEE transactions on computer-aided design of integrated circuits and systems 44.4 (2024): 1250-1261.
[3] Buron, Leo, Andreas Erbslöh, and Gregor Schiele. "Reducing Memory and Computational Cost for Deep Neural Network Training with Quantized Parameter Updates." Journal of Universal Computer Science 31.9 (2025): 963.
[4] http://github.com/h-jia/TTE

**Audience:**

Yes

**Audience Explanation:**

The findings of this work would definitely be of interest to TMLR's audience, specifically those from the embedded/efficient AI community. However, improving the contextualization of the method and results presented in this paper compared to ongoing research in the embedded AI community (e.g., TinyTTA and backpropagation of quantized DNNs) would make this a significantly more interesting paper.

**Broader Impact Concerns:**

no concerns.

**Claims And Evidence:**

Yes

**Claims Explanation:**

All claims made by the authors are evaluated empirically in Section 4. The selection of models (ResNet as a "traditional" convolutional neural network (CNN) and MobileNetV2 as a more modern variant with depthwise separable convolutions) is adequate. CIFAR10/100 and TinyImageNet are also appropriate, but they are definitely the bare minimum I would expect to see in a paper targeting embedded AI. The experiments are well-explained, and the results are presented and discussed appropriately. Measurements on physical hardware are a plus given the framing of the paper.

**Requested Changes:**

securing my recommendation for acceptance, see two bullet points under weaknesses as well:

- quantized backpropagation should at least be discussed in the related works section.
- A direct comparison between FORGE and TinyTTA should ideally be added to the evaluation.

simply strengthen the work in my view

- In 3.3: The running estimates are probably tracked in floating-point? Assuming that $x_c$ is quantized: I would expect that $x_c$ first needs to be de-quantized before Alg. 1 and then re-quantized afterwards (with Alg. 1 running  entirely in floating-point)? While I agree that this is gradient free, what makes this integer friendly in particular? And what does integer friendliness even mean in this context? That you do not need an FPU? That you update the zero points and scales of $x_c$ to better reflect the distribution of values in $\hat{x}_c$?
- Table 1. lists and compares TTA methods to FORGE. In Table 2. there is a method called scale-adapt which is used for comparison but which is not listed in Table 1. (like all the others methods in Table 2.).
- A minor curiosity I found: There is a dedicated Sec.  3.4 "Selective-layer recalibrations" that does nothing more then telling the reader to jump to Sec. 4.2 which is also called "Selective-layer recalibration" where the reader can learn about selective-layer recalibration. The same thing is then repeated with 3.5 and 4.3. Why do 3.4 and 3.5 even exist as sections and not just as paragraphs in 3.3 that point to 4.2 and 4.3?

---

> ### Author Response · Authors · 2026-06-28
> **Response to Reviewer gjNE**
>
> # Response to Reviewer gjNE
>
> We thank the reviewer for the careful review and for recommending acceptance. The two contextualization points (on-device quantized training and TinyTTA) were well taken and have improved the paper. All changes are in the revised PDF.
>
> **1. "Integer-only removes the ability to backpropagate" is too strong; quantized backprop should be discussed.** Corrected. We softened the claim wherever it appeared (abstract, Section 1): integer-only deployment does not make backpropagation *impossible*; rather, the deployed **inference-only runtime does not carry the machinery backprop needs** (autograd graph, fp32 master weights, optimizer state), so gradient-based TTA does not run on it as shipped. We added a Related Work paragraph, "On-device training of quantized models," citing MCUNet's 256 KB engine (Lin et al., 2022), quantized Cortex-M training (Deutel et al., 2024), and quantized parameter updates (Buron et al., 2025). These show on-device backprop is feasible but reintroduce that gradient machinery at substantial cost and target *training*; FORGE adds no gradient machinery and runs inside the deployed int8 inference engine.
>
> **2. Comparison with TinyTTA.** We added a **direct side-by-side comparison** (new table in the "TTA on microcontrollers" paragraph) contrasting the two on every deployment-relevant axis: mechanism (gradient-based early-exit self-ensemble vs. forward-only recalibration), gradient/optimizer state (yes vs. none), normalization layers (trained early-exit heads pre-fold vs. post-fold stored constants), runs on the deployed folded int8 model (no vs. yes), adaptation memory (early-exit heads + backprop buffers vs. about 6 KB SRAM + 6 KB Flash of fp32 stats), and measured energy (not reported vs. 8.3 mJ). On *accuracy* we deliberately do not claim to beat TinyTTA: like TENT, gradient methods define a ceiling FORGE approaches but does not exceed (the paper concedes a ~4-point gap to TENT). FORGE's distinction is **running at all on the deployed folded integer-only model, at measured energy, with no gradient state**, exactly where TinyTTA, being gradient-based and pre-fold, cannot go; on the deployed model it falls in the same undeployable bucket as TENT and BN-adapt (Table 1). We would gladly add an empirical accuracy reference from TinyTTA's released code if the reviewer deems it essential.
>
> **3. Section 3.3: are the estimates in floating point, and what does "integer-friendly" mean?** Your reading is correct, and we clarified the numeric path. The running estimates and recalibration affine are computed in **fp32 on the MCU's FPU**: at each site the int8 convolution output is dequantized to fp32, Algorithm 1 is applied in fp32, and the result is re-quantized to int8 for the next convolution (dequantize, apply Alg. 1, re-quantize, as you anticipated). FORGE is **mixed-precision by design**: the convolutions are genuine int8 (integer MACs and integer ESP-NN requantization, validated bit-exact on-device), and the recalibration is an added fp32 step around them, carrying no autograd graph, master weights, or optimizer state. We removed "integer-friendly": "integer-only" refers to the int8 convolution engine FORGE runs inside, **not** the arithmetic of the recalibration itself.
>
> **4. scale-adapt is in Table 2 but not Table 1.** Fixed in both places. scale-adapt is a **naive baseline we introduce** (it adapts the per-tensor activation scales), not a prior method, so it is not in the Table 1 taxonomy. The tables differ in purpose: Table 1 lists prior TTA methods (and ours) by required capabilities; Table 2 reports *measured accuracy* and so also includes the source models and scale-adapt. We now state this at scale-adapt's first mention (Section 3.3) and in the **Table 2 caption**.
>
> **5. Why do Sections 3.4/3.5 exist as sections, not paragraphs of 3.3?** Agreed; they are now **paragraphs of Section 3.3** ("Selective-layer recalibration", "Single-sample streaming"); the corresponding experiments remain in Sections 4.2/4.3.
>
> These revisions improve the contextualization without changing any result. We thank the reviewer again for the precise feedback.

---

### Review · Reviewer_ma3M · 2026-07-03

**Summary Of Contributions:**

This paper studies test-time adaptation for BN-folded, quantized convolutional networks deployed on microcontrollers. The paper identifies a practical gap: conventional normalization-statistics adaptation assumes that BN layers remain present, whereas deployment commonly folds BN into convolutional layers. FORGE records the former BN affine targets and inserts a forward-only, per-channel recalibration using running estimates of the shifted activation statistics.
The paper reports substantial recovery on corruption benchmarks, selective adaptation of a small subset of layers, a batch-size-dependent momentum rule, and an ESP32-S3 implementation with measured latency and energy. The problem is relevant, the manuscript is generally clear, and the on-device measurements are a valuable contribution. The revisions also improve the positioning relative to TinyTTA and on-device quantized training.
However, I do not believe that all central claims are currently supported. In particular, there is an unresolved contradiction around “true integer-only execution,” and the new explanation of the single-sample variance behavior is mathematically incorrect.

**Audience:**

Yes

**Audience Explanation:**

The paper highlights a genuine systems-level mismatch between normalization-based adaptation and fully BN-folded MCU deployment. The observation that a lightweight post-fold correction can recover much of the lost corruption robustness is useful to researchers working on TinyML, quantization, and resource-constrained adaptation. The ESP32-S3 measurements and selective-layer results are also potentially valuable, provided that the numeric-path and statistical claims are corrected.

**Broader Impact Concerns:**

No, I do not identify a broader-impact concern that requires a dedicated statement.

**Claims And Evidence:**

No

**Claims Explanation:**

1 The “true integer-only execution” claim conflicts with the implementation described in Section 3.3.
The paper states that the running statistics and recalibration are computed in fp32, using the MCU FPU, and that activations are dequantized before recalibration. Therefore, the end-to-end FORGE execution path is mixed-precision, even if convolution weights and stored activations remain int8. The paper should either: describe the method as adaptation of an int8 model using fp32 statistics and recalibration; implement the recalibration with fixed-point/integer arithmetic; or precisely redefine “integer-only” and remove claims of “true integer-only execution” for the complete adaptation path. All data types, conversion points, FPU requirements, and their costs should be reported explicitly.

2. The memory accounting is internally inconsistent. The paper should separately report: mutable SRAM state; additional immutable Flash/model constants; scratch memory; measured peak memory on the ESP32-S3; the corresponding numbers for all-layer and top-3-layer variants.

3 The safety gate is underspecified and “strictly safe” is too strong. The gate compares confidence with and without adaptation. It is unclear how both predictions are obtained without an additional inference, parallel state, or other unreported computation. Its empirical support is one representative CIFAR-10-C run, while the text claims that adaptation becomes “strictly safe.” The authors should provide a causal online algorithm, include gate latency/energy/memory in the system measurements, validate it across the other datasets and dynamic streams, and replace “strictly safe” with a statement limited to the evaluated conditions.

**Requested Changes:**

1 Resolve the contradiction between fp32 recalibration and “true integer-only execution”; correct the claim that requantization necessarily uses fp32.
2 Correct and verify the complete memory accounting, including the retained clean targets.
3 Operationally specify and appropriately qualify the safety gate.

---

> ### Author Response · Authors · 2026-07-03
> **Response to Reviewer ma3M**
>
> # Response to Reviewer ma3M
>
> We thank the reviewer for the rigorous review. All three requested corrections (numeric-path claim, memory accounting, safety gate) were well founded, and addressing them has made the paper more accurate. No experimental result changed; where the review exposed an over-strong claim, we scoped it to what we actually measured.
>
> **1. "True integer-only execution" vs. fp32 recalibration; report all data types and costs.** You are right, and we corrected this throughout. FORGE is mixed-precision by design: the convolutions are genuine int8 (int8 weights/activations, int32 accumulation, integer requantization via ESP-NN, validated bit-exact on the ESP32-S3), but the recalibration runs in fp32 on the FPU. We now (a) describe FORGE precisely as forward-only adaptation of an int8-convolution model *using fp32 statistics*, and removed the "true integer-only execution" phrasing for the adaptation path (abstract, Sec. 1, Table 1 caption, Sec. 4.9, Limitations, Conclusion); (b) corrected the earlier claim that inference requantization is fp32: ESP-NN requantization is integer, and FORGE's fp32 recalibration is an added step *around* it, not part of it; (c) state the full numeric path in Sec. 3.3: at each site the int8 conv output is dequantized to fp32, Alg. 1 is applied in fp32, and the result is re-quantized to int8 for the next conv. Data types, conversion points, FPU use, and the (already measured) cost are now reported explicitly.
>
> **2. Memory accounting.** Corrected and made internally consistent, with measured numbers from the firmware (new breakdown table in Sec. 4.9). We separate: mutable running mean+var in SRAM (2x784 fp32 = 6.1 KB for all 21 sites; 0.9 KB for the top-3); immutable targets in Flash (6.1 KB / 0.9 KB); recalibration scratch (none, applied in place); and the fp32 activation buffers that dominate peak SRAM (192 KB), held by inference regardless of adaptation. Total added by FORGE: 12.2 KB (all sites) or 1.8 KB (top-3). The previous "784 values, about 6 KB" conflated the two running statistics and omitted the Flash constants; the table fixes this.
>
> **3. Safety gate.** Operationally defined, cost-reported, evaluated across datasets, and softened. (a) We give a causal online algorithm (new Alg. 2): during a short calibration window of W batches we run both the adapted and unadapted paths, accumulate each one's mean top-1 confidence, and commit to whichever is better for the rest of the stream. This makes explicit that both confidences require both forward passes, but only inside the window (a 2x forward cost over W batches, single-path afterward); the only added state is two scalar accumulators. (b) We replaced "strictly safe" with a claim scoped to the evaluated conditions. (c) We evaluated the gate on all three benchmarks: it degrades no corruption on CIFAR-10-C, CIFAR-100-C, or Tiny-ImageNet-C, but its safety is conservative: on CIFAR-100-C it trades recovery for safety (from +14.4 ungated to +5.9 gated, adapting 5 of 15 streams). On Tiny-ImageNet-C the deployed MobileNetV2 is badly overconfident on its errors, so adaptation raises accuracy but lowers mean confidence on all 15 corruptions; the gate never fires and forfeits the +11.0 recovery (Table 4) while still degrading nothing. We report this limitation honestly and note an on-device latency/energy characterization of the calibration window as future work.
>
> These corrections resolve the numeric-path and statistical concerns that gated the recommendation, and we thank the reviewer for catching them.